# OPTIMAL MARGIN DISTRIBUTION NETWORK

## ABSTRACT

Recent research about margin theory has proved that maximizing the minimum margin like support vector machines does not necessarily lead to better performance, and instead, it is crucial to optimize the margin distribution. In the meantime, margin theory has been used to explain the empirical success of deep network in recent studies. In this paper, we present ODN (the Optimal margin Distribution Network), a network which embeds a loss function in regard to the optimal margin distribution. We give a theoretical analysis for our method using the PAC-Bayesian framework, which confirms the significance of the margin distribution for classification within the framework of deep networks. In addition, empirical results show that the ODN model always outperforms the baseline cross-entropy loss model consistently across different regularization situations. And our ODN model also outperforms the cross-entropy loss (Xent), hinge loss and soft hinge loss model in generalization task through limited training data.

## 1 INTRODUCTION

In the history of machine learning research, the large margin principle has played an important role in theoretical analysis of generalization ability, meanwhile, it also achieves remarkable practical results for classification (Cortes & Vapnik, 1995) and regression problems (Drucker et al., 1997). More than that, this powerful principle has been used to explain the empirical success of deep neural network. Bartlett et al. (2017) and Neyshabur et al. (2018) present a margin-based multi-class generalization bound for neural networks that scales with their margin-normalized spectral complexity using two different proving tools. Moreover, Arora et al. (2018) proposes a stronger generalization bounds for deep networks via a compression approach, which are orders of magnitude better in practice.

As for margin theory, Schapire et al. (1997) first introduces it to explain the phenomenon that AdaBoost seems resistant to overfitting problem. Two years later, Breiman (1999) indicates that the minimum margin is important to achieve a good performance. However, Reyzin & Schapire (2006) conjectures that the margin distribution, rather than the minimum margin, plays a key role in being empirically resistant to overfitting problem; this has been finally proved by Gao & Zhou (2013). In order to restrict the complexity of hypothesis space suitably, a possible way is to design a classifier to obtain optimal margin distribution.

Gao & Zhou (2013) proves that, to attain the optimal margin distribution, it is crucial to consider not only the margin mean but also the margin variance. Inspired by this idea, Zhang & Zhou (2016) proposes the optimal margin distribution machine (ODM) for binary classification, which optimizes the margin distribution through the first- and second-order statistics, i.e., maximizing the margin mean and minimizing the margin variance simultaneously. To expand this method to the multi-class classification problem, Zhang & Zhou (2017) presents a multi-class version of ODM.

Based on these recent works, we consider the expansion of the optimal margin distribution principle on deep neural networks. In this paper, we propose an optimal margin distribution loss for convolution neural networks, which is not only maximizing the margin mean but also minimizing the margin variance as ODM does. Moreover, we use the PAC-Bayesian framework to derive a novel generalization bound based on margin distribution. Comparing to the spectrally-normalized margin bounds of Bartlett et al. (2017) and Neyshabur et al. (2018), our generalization bound shows that we can restrict the capacity of the model by setting an appropriate ratio between the first-order statistic and the second-order statistic rather than trying to control the whole product of the spectral norms of each layer. And we empirically evaluate our loss function on deep network across different datasets

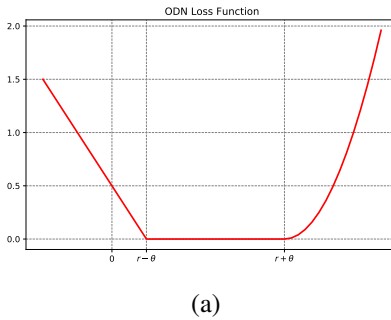 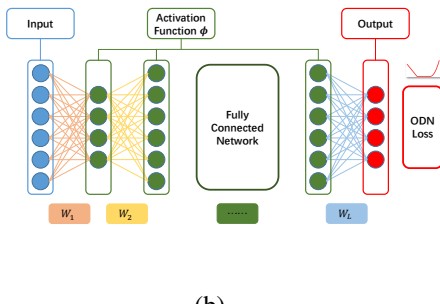

(a)  (b)

Figure 1: The optimal margin distribution loss function (a) and the structure of fully connected network with optimal margin distribution loss (b).

and model structures. Specifically, we consider the performance of these models in generalization task through limited training data.

Recently, many researchers try to explain the experimental success of deep neural network. One of the research direction is to explain why the deep learning does not have serious overfitting problem. Although several common techniques, such as dropout (Srivastava et al., 2014), batch normalization (Ioffe & Szegedy, 2015), and weight decay (Krogh & Hertz, 1992), do improve the generalization performance of the over-parameterized deep models, these techniques do not have a solid theoretical foundation to explain the corresponding effects. As for our optimal margin distribution loss, it has a generalization bound to prove that we can restrict the complexity of hypothesis space reasonably through searching appropriate statistics dependent on data distribution. In experimental section, we compare our optimal margin distribution loss with the baseline cross-entropy loss under different regularization methods.

## 2  OPTIMAL MARGIN DISTRIBUTION LOSS

Consider the classification problem with input domain $\mathcal{X} = \{\boldsymbol{x}|\ \boldsymbol{x} \in \mathbb{R}^n\}$ and output domain $\mathcal{Y} = \{1, \ldots, k\}$, we denote a labeled sample as $\boldsymbol{z} \in (\mathcal{X}, \mathcal{Y})$. Suppose we use a network generating a prediction score for the input vector $\boldsymbol{x} \in \mathcal{X}$ to class $i$, through a function $f_i : \mathcal{X} \to \mathbb{R}$, for $i = 1, \ldots, k$. The predicted label is chosen by the class with maximal score, i.e. $h(x) = \arg\max_i f_i(\boldsymbol{x})$.

Define the decision boundary of each class pair $\{i, j\}$ as:

$$\mathbb{D}_{i,j} := \{\boldsymbol{x}|f_i(\boldsymbol{x}) = f_j(\boldsymbol{x})\}$$

Constructed on this definition, the margin distance of a sample point $\boldsymbol{x}$ to the decision boundary $\mathbb{D}_{i,j}$ is defined by the smallest translation of the sample point to establish the equation as:

$$|\gamma(\boldsymbol{x}, i, j)| = \min_{\boldsymbol{\delta}} \|\boldsymbol{\delta}\|_2 \quad \text{s.t.} \quad f_i(\boldsymbol{x} + \boldsymbol{\delta}) = f_j(\boldsymbol{x} + \boldsymbol{\delta}).$$

In order to approximate the margin distance in the nonlinear situation, Elsayed et al. (2018) has offered a linearizing definition:

$$|\widehat{\gamma}(\boldsymbol{x}, i, j)| = \frac{|f_i(\boldsymbol{x}) - f_j(\boldsymbol{x})|}{\|\nabla_{\boldsymbol{x}} f_i(\boldsymbol{x}) - \nabla_{\boldsymbol{x}} f_j(\boldsymbol{x})\|_2}.$$

Naturally, this pairwise margin distance leads us to the following definition of the margin for a labeled sample $\boldsymbol{z} = (\boldsymbol{x}, y)$:

$$\gamma_h(\boldsymbol{x}, y) = \frac{f_y(\boldsymbol{x}) - \max_{i \neq y} f_i(\boldsymbol{x})}{\|\nabla_{\boldsymbol{x}} f_y(\boldsymbol{x}) - \nabla_{\boldsymbol{x}} \max_{i \neq y} f_i(\boldsymbol{x})\|_2}.$$

Therefore, the defined classifier $h$ misclassifies $(\boldsymbol{x}, y)$ if and only if the margin is negative. Given a hypothesis space $\mathcal{H}_{\mathbb{S}}$ of functions mapping $\mathcal{X}$ to $\mathcal{Y}$, which can be learned by the fixed deep neural

network through the training set $\mathbb{S}$, our purpose is to find a way to learn a decision function $h \in \mathcal{H}_{\mathbb{S}}$ such that the generalization error $R(h) = \mathbb{E}_{(\boldsymbol{x}, y) \sim (\mathcal{X}, \mathcal{Y})}[\mathbf{1}_{h(\mathrm{x}) \neq \mathrm{y}}]$ is small.

In this work, we introduce a type of margin loss, and connect it to deep neural networks. The origin loss function has been specially adapted for the difference between deep learning models and linear models by us as following definition:

$$\ell_{r, \theta, \mu}(\gamma_h) = \begin{cases} \frac{r - \theta - \gamma_h}{r - \theta} & \gamma_h \leq r - \theta \\ 0 & r - \theta < \gamma_h \leq r + \theta \\ \frac{\mu(\gamma_h - r - \theta)^2}{(r + \theta)^2} & \gamma_h > r + \theta, \end{cases} \tag{1}$$

where $r$ is the margin mean, $\theta$ is the margin variance and $\mu$ is a parameter to trade off two different kinds of deviation (keeping the balance on both sides of the margin mean). In the Appendix A, we explain the reason why we use these three hyper-parameters to construct such a optimal margin distribution loss function.

Fig. 1 shows, equation 1 will produce a linear loss decreasing progressively when the margins of sample points satisfy $\gamma_h \leq r - \theta$ and a square loss increasing progressively when the margins satisfy $\gamma_h \geq r + \theta$. Therefore, our margin loss function will enforce the tie which has zero loss to contain the sample points as many as possible. So the parameters of the classifier will be determined not only by the samples that are close to the decision boundary but also by the samples that are away from the decision boundary. In other words, our loss function is aimed at finding a decision boundary which is determined by the whole sample margin distribution, instead of the minority samples that have minimum margins. To verify superiority of the optimal margin distribution network, our paper verifies it both theoretically and empirically.

## 3 ANALYSIS

To present a new margin bound for our optimal margin distribution loss, some notations are needed. Consider that the convolution neural networks can be regarded as a special structure of the fully connected neural networks, we simplify the definition of the deep networks. Let $f_{\boldsymbol{w}}(\boldsymbol{x}) : \mathcal{X} \to \mathbb{R}^k$ be the function learned by a L-layer feed-forward network for the classification task with parameters $\boldsymbol{w} = (\boldsymbol{W}_1, \ldots, \boldsymbol{W}_L)$, thus $f_{\boldsymbol{w}}(\boldsymbol{x}) = \boldsymbol{W}_L \phi(\boldsymbol{W}_{L-1} \phi(\ldots \phi(\boldsymbol{W}_1 \boldsymbol{x})))$, here $\phi$ is the ReLU activation function. Let $f_{\boldsymbol{w}}^i$ denote the output of layer $i$ before activation and $\rho$ be an upper bound on the number of output units in each layer. Recursively, we can redefine the deep network: $f_{\boldsymbol{w}}^1(\boldsymbol{x}) = \boldsymbol{W}_1 \boldsymbol{x}$ and $f_{\boldsymbol{w}}^i(\boldsymbol{x}) = \boldsymbol{W}_i \phi(f_{\boldsymbol{w}}^{i-1}(\boldsymbol{x}))$. Let $\|\cdot\|_F$, $\|\cdot\|_1$ and $\|\cdot\|_2$ denote the Frobenius norm, the element-wise $\ell_1$ norm and the spectral norm respectively.

In order to facilitate the theoretical derivation of our formula, we simplify the definition of the loss function:

$$\begin{aligned} L_{r, \theta}(f_{\boldsymbol{w}}) = & \Pr_{(\boldsymbol{x}, y) \sim (\mathcal{X}, \mathcal{Y})} \left[ \frac{f_{\boldsymbol{w}, y}(\boldsymbol{x}) - \max_{i \neq y} f_{\boldsymbol{w}, i}(\boldsymbol{x})}{\nabla_{\boldsymbol{x}} f_{\boldsymbol{w}, y}(\boldsymbol{x}) - \nabla_{\boldsymbol{x}} \max_{i \neq y} f_{\boldsymbol{w}, i}(\boldsymbol{x})} \leq r - \theta \right] \\ & + \Pr_{(\boldsymbol{x}, y) \sim (\mathcal{X}, \mathcal{Y})} \left[ \frac{f_{\boldsymbol{w}, y}(\boldsymbol{x}) - \max_{j \neq y} f_{\boldsymbol{w}, j}(\boldsymbol{x})}{\nabla_{\boldsymbol{x}} f_{\boldsymbol{w}, y}(\boldsymbol{x}) - \nabla_{\boldsymbol{x}} \max_{j \neq y} f_{\boldsymbol{w}, j}(\boldsymbol{x})} \geq r + \theta \right]. \end{aligned}$$

Specially, define the $L_0$ as $r = \theta$ and $\theta \to \infty$, actually equal to the 0-1 loss. And let $\widehat{L}_{r, \theta}(f_{\boldsymbol{w}})$ be the empirical estimate of the optimal margin distribution loss. So we will denote the expected risk and the empirical risk as $L_0(f_{\boldsymbol{w}})$ and $\widehat{L}_0(f_{\boldsymbol{w}})$, which are bounded between 0 and 1.

In the PAC-Bayesian framework, one expresses the prior knowledge by defining a prior distribution over the hypothesis class. Following the Bayesian reasoning approach, the output of the learning algorithm is not necessarily a single hypothesis. Instead, the learning process defines a posterior probability over $\mathcal{H}$, which we denote by $Q$. In the context of a supervised learning problem, where $\mathcal{H}$ contains functions from $\mathcal{X}$ to $\mathcal{Y}$, one can think of $Q$ as defining a randomized prediction rule. We consider the distribution $Q$ which is learned from the training data of form $f_{\boldsymbol{w} + \boldsymbol{u}}$, where $\boldsymbol{u}$ is a random variable whose distribution may also depend on the training data. Let $P$ be a prior distribution over $\mathcal{H}$ that is independent of the training data, the PAC-Bayesian theorem states that with possibility at least $1 - \delta$ over the choice of an i.i.d. training set $\mathbb{S} = \{\boldsymbol{z}_1, ..., \boldsymbol{z}_m\}$ sampled according to $(\mathcal{X}, \mathcal{Y})$, for all distributions $Q$ over $\mathcal{H}$ (even such that depend on $S$), we have (McAllester, 2003):

$$\mathbb{E}_{\boldsymbol{u}}[L_0(f_{\boldsymbol{w}+\boldsymbol{u}})] \le \mathbb{E}_{\boldsymbol{u}}[\widehat{L}_0(f_{\boldsymbol{w}+\boldsymbol{u}})] + \sqrt{\frac{\left(D_{\mathrm{KL}}(\boldsymbol{w}+\boldsymbol{u}\|P) + \ln\frac{m}{\delta}\right)}{2(m-1)}}. \tag{2}$$

Note that the left side of the inequality is based on $f_{\boldsymbol{w}+\boldsymbol{u}}$. To derive an expected risk bound $L_0(f_{\boldsymbol{w}})$ for a single predictor $f_{\boldsymbol{w}}$, we have to relate this PAC-Bayesian bound to the expected perturbed loss just like Neyshabur et al. (2018) derive the Lemma 1 in their paper. Based on the inequality 2, we introduce a perturbed restriction which is related to the margin distribution (the margin mean $r$ and margin variance $\theta$):

**Lemma 1.** *Let $f_{\boldsymbol{w}}(\boldsymbol{x}) : \mathcal{X} \to \mathbb{R}^k$ be any predictor with parameters $\boldsymbol{w}$, and $P$ be any distribution on the parameters that is independent of the training data. Then, for any $r > \theta > 0$, $\delta > 0$, with probability at least $1 - \delta$ over the training set of size $m$, for any $\boldsymbol{w}$, and any random perturbation $\boldsymbol{u}$ s.t. $\mathrm{Pr}_{\boldsymbol{u}}\left[\max_{\boldsymbol{x}\in\mathcal{X}}|f_{\boldsymbol{w}+\boldsymbol{u}}(\boldsymbol{x}) - f_{\boldsymbol{w}}(\boldsymbol{x})|_{\infty} < \frac{r-\theta}{4}\right] \ge \frac{1}{2}$, we have:*

$$L_0(f_{\boldsymbol{w}}) \le \widehat{L}_{r,\theta}(f_{\boldsymbol{w}}) + \sqrt{\frac{D_{\mathrm{KL}}(\boldsymbol{w}+\boldsymbol{u}\|P) + \ln\frac{3m}{\delta}}{m-1}}.$$

The margin variance information does not change the conclusion of the perturbed restriction, the proof of this lemma is similar to Lemma 1 in Neyshabur et al. (2018).

In order to bound the change caused by perturbation, we have to bring in three definitions that are used to formalize error-resilience in Arora et al. (2018) as follows:

**Definition 1.** *(Layer Cushion). The layer cushion of layer $i$ is defined to be largest number $\mu_i$ such that for any $\boldsymbol{x} \in \mathbb{S}$:*

$$\mu_i\|\boldsymbol{W}_i\|_F\|\phi(f_{\boldsymbol{w}}^{i-1}(\boldsymbol{x}))\|_2 \le \|f_{\boldsymbol{w}}^i(\boldsymbol{x})\|_2.$$

Intuitively, cushion considers how much smaller the output $\boldsymbol{W}_i\phi(f_{\boldsymbol{w}}^{i-1}(\boldsymbol{x}))$ is compared to the upper bound $\|\boldsymbol{W}_i\|_2\|\phi f_{\boldsymbol{w}}^{i-1}(\boldsymbol{x})\|_2$. However, for nonlinear operators the definition of error resilience is less clean. Let's denote $M^{i,j} : \mathbb{R}^{h^i} \to \mathbb{R}^{h^j}$ the operator corresponding to the portion of the deep network from layer $i$ to layer $j$, and by $J^{i,j}$ its Jacobian. If infinitesimal noise is injected before level $i$ then $M^{i,j}$ passes it like $J^{i,j}$, a linear operator. When the noise is small but not infinitesimal then one hopes that we can still capture the local linear approximation of the nonlinear operator $M$ by define Interlayer Cushion:

**Definition 2.** *(Interlayer Cushion). For any two layers $i < j$, we define the interlayer cushion $\mu_{i,j}$, as the largest number such that for any $\boldsymbol{x} \in \mathbb{S}$:*

$$\mu_{i,j}\|J_{f_{\boldsymbol{w}}^i(\boldsymbol{x})}^{i,j}\|_F\|\phi(f_{\boldsymbol{w}}^{i-1}(\boldsymbol{x}))\|_2 \le \|f_{\boldsymbol{w}}^j(\boldsymbol{x})\|_2.$$

*Furthermore, for any layer $i$ we define the minimal interlayer cushion as $\mu_{i\to} = \min_{i\le j\le L}\mu_{i,j} = \min\{\frac{1}{\sqrt{\rho}}, \min_{i\le j\le L}\mu_{i,j}\}$.*

The next condition qualifies a common appearance: if the input to the activations is well-distributed and the activations do not correlate with the magnitude of the input, then one would expect that on average, the effect of applying activations at any layer is to decrease the norm of the pre-activation vector by at most some small constant factor.

**Definition 3.** *(Activation Contraction). The activation contraction $c$ is defined as the smallest number such that for any layer $i$ and any $\boldsymbol{x} \in \mathcal{X}$:*

$$\|\phi(f_{\boldsymbol{w}}^i(\boldsymbol{x}))\|_2 \ge \frac{\|f_{\boldsymbol{w}}^i(\boldsymbol{x})\|_2}{c}$$

To guarantee that the perturbation of the random variable $\boldsymbol{u}$ will not cause a large change on the output with high possibility, we need a perturbation bound to relate the change of output to the structure of the network and the prior distribution $P$ over $\mathcal{H}$. Fortunately, Neyshabur et al. (2018) proved a restriction on the change of the output by norms of the parameter weights. In the following lemma, we preset our hyper-parameters $r$ and $\theta$, s.t. the parameter weights $\boldsymbol{w} \in \mathcal{H}$ satisfying $\|f_{\boldsymbol{w}}(\boldsymbol{x})\|_2 \le r + \theta$, when fixing $\|\boldsymbol{W}_L\|_2 = 1$. Thus, we can bound this change in terms of the spectral norm of the layer and the presetting hyper-parameters:

**Lemma 2.** *For any $L > 0$, let $f_{\boldsymbol{w}} : \mathcal{X} \to \mathbb{R}^k$ be a L-layer network. Then for any $\boldsymbol{w} \in \mathcal{H}$ satisfying $\|f_{\boldsymbol{w}}(\boldsymbol{x})\|_2 \leq r + \theta$, and $\boldsymbol{x} \in \mathcal{X}$, and any perturbation $\boldsymbol{u} = vec(\{\boldsymbol{U}_i\}_{i=1}^L)$ s.t. $\|\boldsymbol{U}_i\|_2 \leq \frac{1}{L}\|\boldsymbol{W}_i\|_2$, the change of the output of the network can be bounded as follows:*

$$|f_{\boldsymbol{w}+\boldsymbol{u}}(\boldsymbol{x}) - f_{\boldsymbol{w}}(\boldsymbol{x})|_2 \leq \mathcal{O}\left(c(r + \theta)\sum_{i=1}^L \frac{\|\boldsymbol{U}_i\|_2}{\|\boldsymbol{W}_i\|_2 \mu_i \mu_{i\to}}\right).$$

The proof of this lemma is given in Appendix B. Eventually, we utilize all above bounding lemmas and the error-resilience definitions to derive the following new margin based generalization bound for our Optimal margin Distribution Network.

**Theorem 1.** *(Generalization Bound). For any $L, \rho > 0$, let $f_{\boldsymbol{w}} : \mathcal{X} \to \mathbb{R}^k$ be a L-layer feed-forward network with ReLU activations. Then, for any $\delta > 0, r > \theta > 0$, with probability $\geq 1 - \delta$ over a training set of size $m$, for any $\boldsymbol{w}$, we have:*

$$L_0(f_{\boldsymbol{w}}) \leq \widehat{L}_{r,\theta}(f_{\boldsymbol{w}}) + \mathcal{O}\left(\sqrt{\frac{c^2 L^2 \rho \ln(L\rho)(r+\theta)^2 \sum_{i=1}^L \frac{\|\boldsymbol{W}_i\|_F^2}{\|\boldsymbol{W}_i\|_2^2 \mu_i \mu_{i\to}} + \ln\frac{Lm}{\delta}}{m(r-\theta)^2}}\right).$$

The proof of this theorem is given in Appendix B.

**Remark.** *Comparing with the spectral complexity in Bartlett et al. (2017):*

$$R_{\boldsymbol{w}} := \left(\prod_{i=1}^L \|\boldsymbol{W}_i\|_2\right)\left(\sum_{i=1}^L \frac{\|\boldsymbol{W}_i - \boldsymbol{I}_i\|_{2,1}^{2/3}}{\|\boldsymbol{W}_i\|_2^{2/3}}\right)^{3/2}, \tag{3}$$

*which is dominated by the product of spectral norms across all, our margin bound is relevant to $r, \theta$ dependent on the margin distribution and $\mu_i$ and $\mu_{i\to}$ dependent on the network structure. The product value in equation 3 is extremely large and is hard to control it, but the parameter in our generalization bound is easy to restrict. Explicitly, the factor consisted of hyper-parameters $\left(\frac{r+\theta}{r-\theta}\right)^2 = \left(\frac{1+\frac{\theta}{r}}{1-\frac{\theta}{r}}\right)^2 = \left(\frac{2}{1-\frac{\theta}{r}} - 1\right)^2$ is a monotonicity increasing function with regard to the ratio $\frac{\theta}{r} \in [0, 1)$. Under the assumption of separability, we can come to the conclusion that maller $\theta$ and larger $r$ make the complexity smaller. Searching a suitable value of $r$ and $\theta$ for the specific data distribution will lead us to a better generalization performance.*

## 4 EXPERIMENT

In this section, we empirically evaluate the effectiveness of our optimal margin distribution loss on generalization tasks, comparing it with three other loss functions: cross-entropy loss (Xent), hinge loss, and soft hinge loss. We first compare them under limited training data situation, using only part of the MNIST dataset (LeCun et al., 1998) to train and evaluate the models deploying the four different losses, with the used data ratio ranging from 0.125% to 100%. Similar experiments are also performed on the legend CIFAR-10 dataset (Krizhevsky & Hinton, 2009). Then we compare them under different regularization situations, investigating the combination of optimal margin distribution loss with dropout and batch normalization. Finally, we visualize and compare the features learned by the deep learning model with the four lose functions as well as the margin distribution from those models.

Here we introduce three commonly used loss functions in deep learning for comparison in the experimental section:

**Cross-entropy Loss (Xent):**

$$L_{Xent}(f_{\boldsymbol{w}}) = -\ln\left(\frac{\exp(f_{\boldsymbol{w},y}(\boldsymbol{x}))}{\sum_{i=1}^k \exp(f_{\boldsymbol{w},i}(\boldsymbol{x}))}\right)$$

**Hinge Loss:**

$$\gamma_h(\boldsymbol{x}, y) = \frac{f_{\boldsymbol{w},y}(\boldsymbol{x}) - \max_{i\neq y} f_{\boldsymbol{w},i}(\boldsymbol{x})}{\|\nabla_{\boldsymbol{x}} f_{\boldsymbol{w},y}(\boldsymbol{x}) - \nabla_{\boldsymbol{x}} \max_{i\neq y} f_{\boldsymbol{w},i}(\boldsymbol{x})\|_2}, \qquad L_{Hinge}(f_{\boldsymbol{w}}) = \max\left[0, \gamma_h(\boldsymbol{x}, y) - \gamma_0\right],$$

where $\gamma_0$ is a hyper-parameter to control the minimum margin as support vector machine does.

**Soft Hinge Loss:**

$$L_{SoftHinge}(f_{\boldsymbol{w}}) = -\frac{1}{k} \cdot \left[ \ln(1 + \exp(-f_{\boldsymbol{w},y}(\boldsymbol{x}))) - \sum_{i \neq y} \ln(1 + \exp(-f_{\boldsymbol{w},i}(\boldsymbol{x}))) \right],$$

where $k$ is the number of classes.

### 4.1 EXPERIMENTAL SETUP

Regarding the deep models, we use the following combination of datasets and models: a simple deep convolutional network for MNIST, original Alexnet (Krizhevsky et al., 2012) for CIFAR-10. In terms of the implementation of optimal margin distribution loss, as shown in Section 2, there is a gradient term in the loss itself, which can make the computation expensive. To reduce computational cost as Elsayed et al. (2018) do, in the backpropagation step we considered the gradient term $\|\nabla_{\boldsymbol{x}} f_y(\boldsymbol{x}) - \nabla_{\boldsymbol{x}} \max_{i \neq y} f_i(\boldsymbol{x})\|_2$ as a constant, so that we recomputed the value of $\|\nabla_{\boldsymbol{x}} f_y(\boldsymbol{x}) - \nabla_{\boldsymbol{x}} \max_{i \neq y} f_i(\boldsymbol{x})\|_2$ at every forward propagation step. Furthermore, since the denominator item could be too small, which would cause numerical problem, we added an $\epsilon$ with small value to the denominator so that clip the loss at some threshold.

For *special hyperparameters*, including the margin mean parameter and margin variance parameter for the ODN model, and margin parameter for hinge loss model, we performed hyperparameter searching. We held out 5000 samples of the training set as a validation set, and used the remaining samples to train models with different special hyperparameters values, on both the MNIST dataset and the CIFAR-10 dataset. As for the common hyperparameters, such as, learning rate and momentum, we set them as the default commonly used values in Pytorch for all the models. We chose batch stochastic gradient descent as the optimizer. Evaluated on the testing dataset, the baseline cross-entropy model achieves a test accuracy of 99.09%; the hinge loss model achieves 98.95% on MNIST dataset; the soft-hinge loss model achieves 99.14% and the ODN model achieves 99.16%. On the CIFAR-10 dataset, the baseline cross-entropy model trained on the remaining training samples achieves a test accuracy of 83.51%; the hinge loss model achieves 82.15%; the soft-hinge loss model achieves 81.96% and the ODN model achieves 84.61%.

### 4.2 LIMITED SMALL SAMPLE LEARNING

It is well-known that deep learning method is very data-hungry, which means that if the training data size decreases, the model's performance can decrease significantly. In reality, this disadvantage of deep learning method can restrict its application seriously since sufficient amount of data is not always available. On the other hand, one of the desirable property of optimal margin distribution loss based models is that it can generalize well even when the training data is insufficient because the optimal margin distribution loss can restrict the complexity of the hypothesis space suitably. To evaluate the performance of optimal margin distribution loss based models under insufficient training data setting, we randomly chose some fraction of the training set, in particular, from 100% of the training samples to 0.125% on the MNIST dataset, and from 100% of the training samples to 0.5% on the CIFAR-10 dataset, and train the models accordingly.

In Fig. 2, we show the test accuracies of cross-entropy, hinge, soft hinge, and optimal margin distribution loss based models trained on different fractions of the MNIST and CIFAR-10 dataset. As shown in the figure, the test accuracies of all these four models increase as the fraction of training samples increases. Obviously, the ODN models proposed by our paper outperform all the other models constantly across different datasets and different fractions. Furthermore, the less training data there are, the larger performance gain the ODN model can have. On the MNIST dataset, the optimal margin distribution loss based model outperforms cross-entropy loss model by around 4.95%, hinge loss model by around 6.84% and soft-hinge loss model by around 3.03% on the smallest training set which contains only 0.125% of the whole training samples. Similarly, The ODN model outperforms cross-entropy loss model by around 9.9%, hinge loss model by around 10.1%, and soft hinge loss model by 13.4% on the smallest CIFAR-10 dataset which contains only 0.5% of the whole training samples.

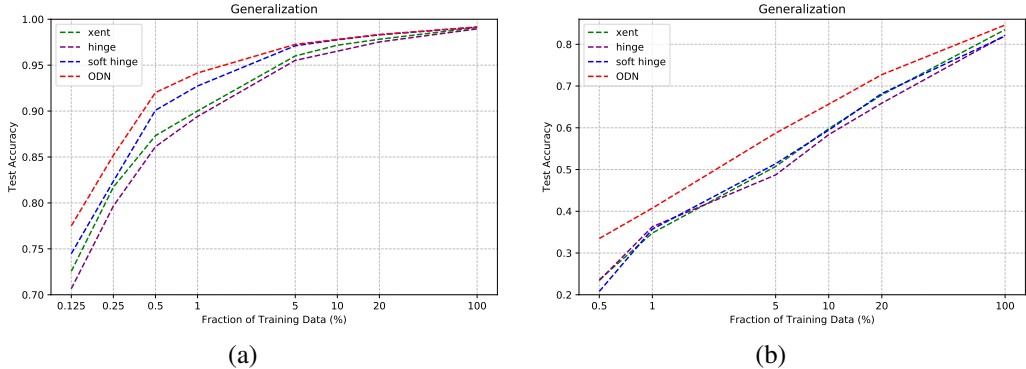

Figure 2: Performance of selected MNIST (a) and CIFAR10 (b) models on generalization tasks.

## 4.3 REGULARIZATION METHODS

Table 1: Test accuracy of Alexnet on CIFAR-10 with different regularization methods and different fraction of training set.

| Accuracy (%) | Batch Normalization | | Non Batch Normalization | |
|---|---|---|---|---|
| | Xent | ODN | Xent | ODN |
| ALL_DROPOUT | $85.782 \pm 0.198$ | $\mathbf{87.644 \pm 0.151}$ | $83.517 \pm 0.322$ | $\mathbf{84.643 \pm 0.255}$ |
| ALL_NON_DROPOUT | $81.491 \pm 0.143$ | $\mathbf{86.233 \pm 0.244}$ | $72.223 \pm 1.284$ | $\mathbf{76.793 \pm 1.279}$ |
| 5%_DROPOUT | $61.955 \pm 1.945$ | $\mathbf{67.636 \pm 1.633}$ | $50.747 \pm 3.735$ | $\mathbf{58.739 \pm 1.348}$ |
| 5%_NON_DROPOUT | $57.753 \pm 2.228$ | $\mathbf{64.173 \pm 1.982}$ | $36.293 \pm 4.872$ | $\mathbf{47.056 \pm 3.927}$ |
| Hyper-param (r / $\theta$ / $\mu$) | - | 30/0.7/0.1 | - | 1.2/0.7/0.1 |

We also compared our optimal margin distribution loss with the baseline cross-entropy loss under different regularization methods and different amounts of training data, whose results are shown in Table 1. As suggested by Table 1, our loss can outperform the baseline loss consistently across different situations, no matter whether dropout, batch normalization or all the CIFAR-10 dataset are used or not. Specifically, when the size scale of training samples is small (5% fraction of the CIFAR-10 training set), the advantage of our optimal margin distribution loss is more significant. Moreover, our optimal margin distribution loss can cooperate with batch normalization and dropout, achieving the best performance in Table.1, which is shown in bold red text. Unlike dropout and batch normalization which are lack of solid theory ground, our optimal margin distribution loss has the margin bound , which guides us to find the suitable ratio $\frac{\theta}{r}$ to restrict the capacity of models and alleviate the overfitting problem efficiently.

## 4.4 DATA VISUALIZATION

Since the performance of the ODN models is excellent, we hope to see that the distributions of data in the learned feature space (the last hidden layer) are consistent with the generalization results. In this experiment, we use t-SNE method to visualize the data distribution on the last hidden layer for training samples and test samples. Fig. 3 and Fig. 4 plots the 2-dimension embedding image on limited MNIST and CIFAR-10 dataset, which is only 1% of the whole training samples. t-SNE (Maaten & Hinton, 2008) is a tool to visualize high-dimensional data. It converts similarities between data points to joint probabilities and tries to minimize the Kullback-Leibler divergence between the joint probabilities of the low-dimensional embedding and the high-dimensional data.

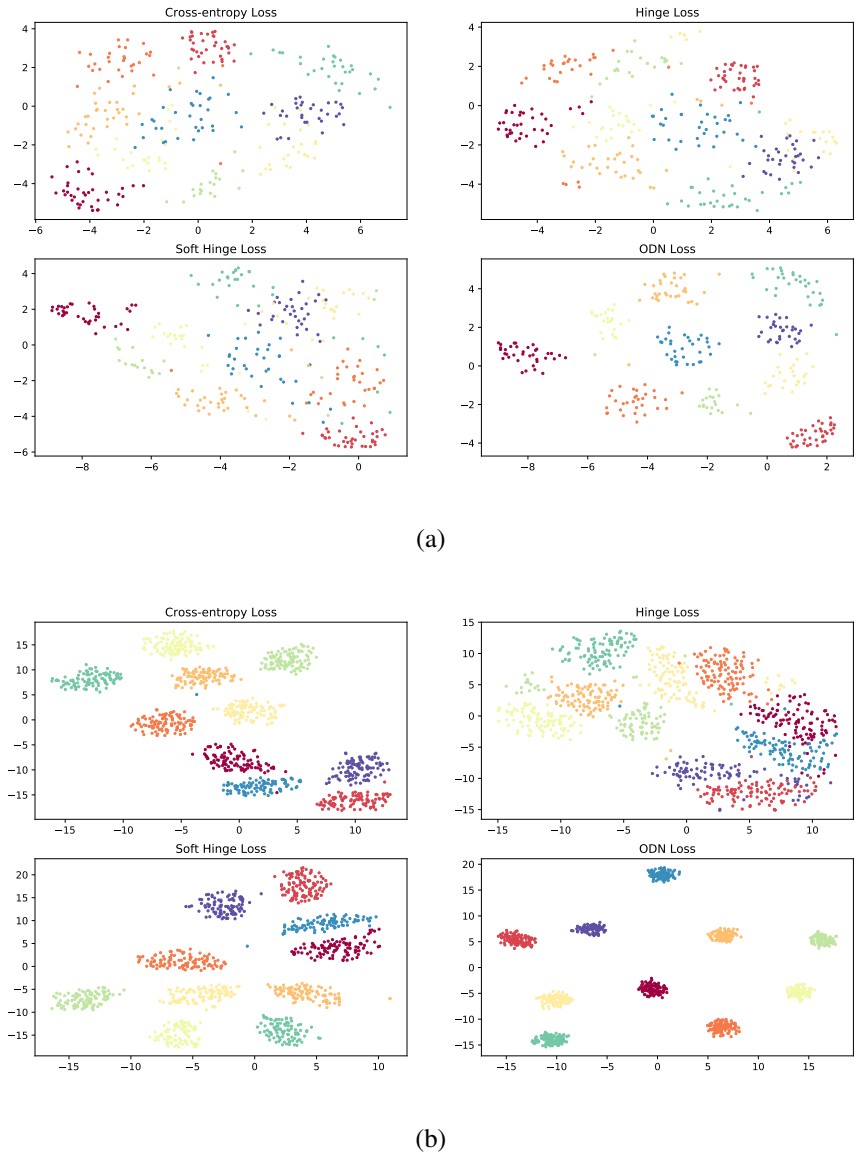

(a)

(b)

Figure 3: Learned features visualization of selected MNIST models on training set (a); Learned features visualization of selected CIFAR-10 models on training set (b).

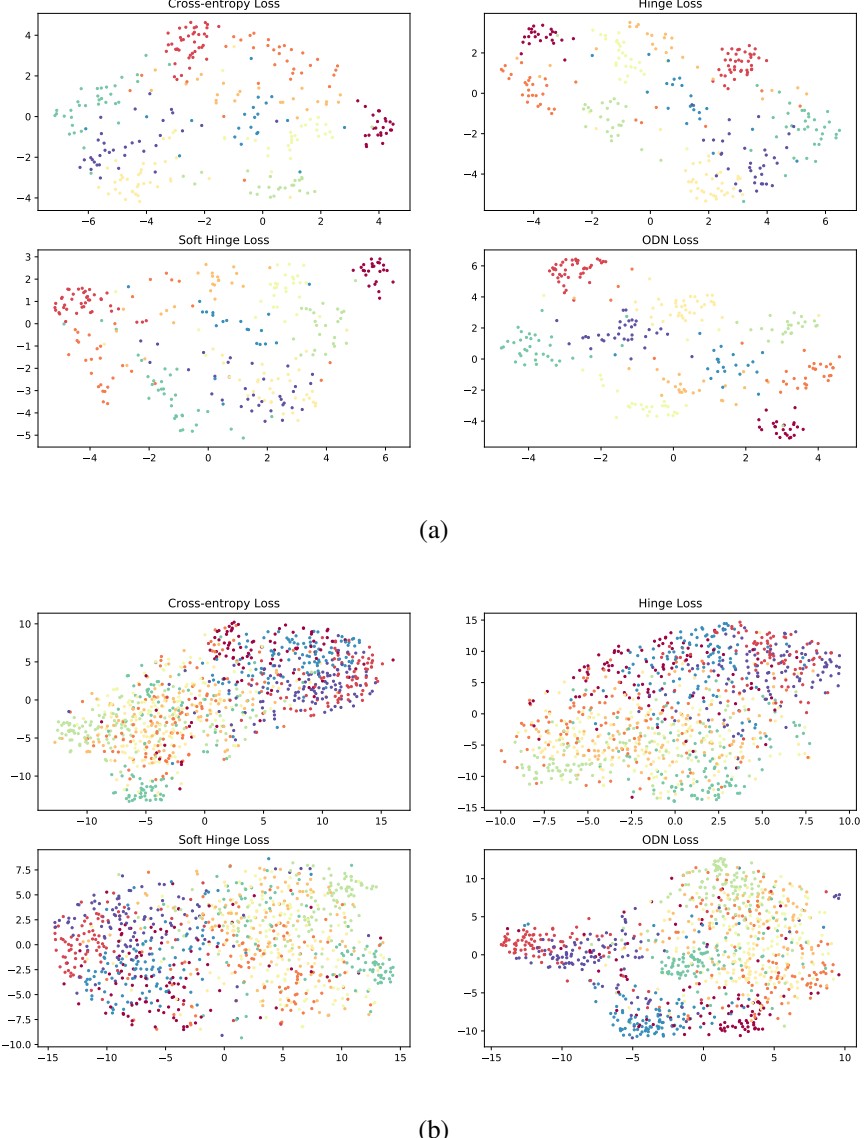

Figure 4: Learned features visualization of selected MNIST models on testing set (a); Learned features visualization of selected CIFAR-10 models on testing set (b).

Table 2: Variance decomposition of selected MNIST models on embedding space.

| Models | Training data | | | | Test data | | | |
|---|---|---|---|---|---|---|---|---|
| | Xent | Hinge | Soft Hinge | ODN | Xent | Hinge | Soft Hinge | ODN |
| *Inter Class Var* | 522 | 529 | 466 | 190 | 831 | 811 | 854 | 649 |
| *Intra Class Var* | 11200 | 11092 | 14128 | 16469 | 13007 | 12986 | 11362 | 13955 |
| *ratio* | 21.45 | 20.96 | 30.32 | **86.68** | 15.65 | 16.01 | 13.3 | **21.5** |

Table 3: Variance decomposition of selected CIFAR-10 models on embedding space.

| Models | Training data | | | | Test data | | | |
|---|---|---|---|---|---|---|---|---|
| | Xent | Hinge | Soft Hinge | ODN | Xent | Hinge | Soft Hinge | ODN |
| *Inter Class Var* | 804 | 713 | 637 | 193 | 1993 | 1429 | 1917 | 1279 |
| *Intra Class Var* | 15692 | 9466 | 17546 | 13273 | 7260 | 4780 | 5810 | 5645 |
| *ratio* | 19.52 | 13.28 | 27.55 | **68.77** | 3.64 | 3.34 | 3.03 | **4.41** |

Consistently, we can find that the result of our ODN model is better than all the others, the distribution of the samples which has the same label is more compact. To quantify the degree of compactness of the distribution, we perform a variance decomposition on the data in the embedding space. By comparing the ratio of the intra-class variance to the inter-class variance in Table 2 and Table 3, we can know that our optimal margin distribution loss alway attain the most compact distribution in these four loss functions.

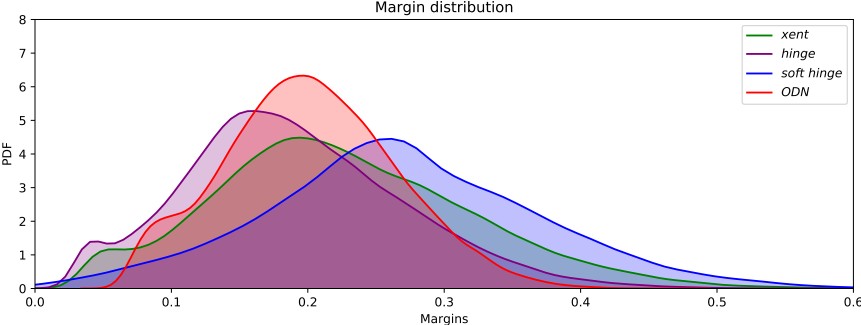

Figure 5: Margin distribution of selected MNIST models.

Moreover, the visualization result is consistent with the margin distribution of these four models in Fig. 5, which means getting an optimal margin distribution is helpful to deriving a good learned features space. And that representation features space can further alleviate the overfitting problem of deep learning. Hence, the optimal margin distribution loss function can significantly outperforms the other loss functions in generalization task through limited training data.

## 4.5 MARGIN DISTRIBUTIONS

Fig. 5 plots the kernel density estimates of margin distribution producted by cross-entropy loss, hinge loss, soft hinge loss and ODN models on dataset MNIST. As can be seen, our ODN model derives a large margin mean with a smallest margin variance in all these four models. By calculating the value of ratio between the margin mean and the margin standard deviation, we know that the ratio in our ODN model is 3.20 which is significantly larger than 2.38 in the cross-entropy loss,

2.35 in the hinge loss and 2.63 in the soft hinge loss. The distribution of our model becomes more "sharper", which prevents the instance with small margin, so our method can still perform well as the training data is limited, which is also consistent with the result in Fig. 2.

## 5  CONCLUSIONS

Recent studies disclose that maximizing the minimum margin for decision boundary does not necessarily lead to better generalization performance, and instead, it is crucial to optimize the margin distribution. However, the influence of margin distribution for deep networks still remains undiscussed. We propose ODN model trying to design a loss function which aims to control the ratio between the margin mean and the margin variance. Moreover, we present a theoretical analysis for our method, which confirms the significance of margin distribution in generalization performance. As for experiments, the results validate the superiority of our method in limited data problem. And our optimal margin distribution loss function can cooperate with batch normalization and dropout, achieving a better generalization performance.

ACKNOWLEDGMENTS

We are grateful to Yu Li and Kangle Zhao for discussions and helpful feedback on the manuscript.

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

## A   EXPLAINATION FOR OPTIMAL MARGIN DISTRIBUTION LOSS

Inspired by the optimal margin distribution principle, Zhang & Zhou (2017) propose the multi-class optimal margin distribution machine, which characterizes the margin distribution according to the first- and second-order statistics. Specially, let $\bar{\gamma}$ denote the margin mean, and the optimal margin distribution machine can be formulated as:

$$
\min_{\boldsymbol{w},\bar{\gamma},\xi_i,\epsilon_i} \Omega(\boldsymbol{w}) - \eta\bar{\gamma} + \frac{\lambda}{m}\sum_{i=1}^{m}(\xi_i^2 + \epsilon_i^2)
$$

$$
\text{s.t.} \quad \gamma_h(\boldsymbol{x}_i, y_i) \geq \bar{\gamma} - \xi_i
$$

$$
\gamma_h(\boldsymbol{x}_i, y_i) \leq \bar{\gamma} + \epsilon_i, \forall i,
$$

where $\Omega(\boldsymbol{w})$ is the regularization term to penalize the norm of the weights, $\eta$ and $\lambda$ are trading-off parameters, $\xi_i$ and $\epsilon$ are the deviation of the margin $\gamma_h(\boldsymbol{x}_i, y_i)$ to the margin mean. It is evident that $\sum_{i=1}^{m}(\xi_i^2 + \epsilon_i^2)/m$ is exactly the margin mean.

In the linear situation, scaling $\boldsymbol{w}$ does not affect the final classification results such as SVM, the margin mean can be normalized as 1, then the deviation of the margin of $(\boldsymbol{x}_i, y_i)$ to the margin mean is $|\gamma_h(\boldsymbol{x}_i, y_i) - 1|$, and the formula can be reconstruct as:

$$
\min_{\boldsymbol{w},\xi_i,\epsilon_i} \Omega(\boldsymbol{w}) + \frac{\lambda}{m}\sum_{i=1}^{m}\frac{\xi_i^2 + \mu\epsilon_i^2}{(1-\theta)^2}
$$

$$
\text{s.t.} \quad \gamma_h(\boldsymbol{x}_i, y_i) \geq 1 - \theta - \xi_i
$$

$$
\gamma_h(\boldsymbol{x}_i, y_i) \leq 1 + \theta + \epsilon_i, \forall i,
$$

where $\mu \in (0, 1]$ is parameter to trade off two different kinds of deviation (keeping the balance on both sides of the margin mean). $\theta \in [0, 1)$ is a parameter of the zero loss band, which can control the number of support vectors. In other words, $\theta$ is a parameter to control the margin variance, while the data which is out of this zero loss band will be used to update the weights to minimize the loss. For this reason, we simply regard it as the margin variance.

However, under the non-linear setting in our paper, we can not directly linearly normalize the margin mean to the value 1. So we assume that the normalized margin mean is $r$, then the optimization target can be reformulated as:

$$
\min_{\boldsymbol{w},\xi_i,\epsilon_i} \Omega(\boldsymbol{w}) + \frac{\lambda}{m}\sum_{i=1}^{m}\frac{\xi_i^2 + \mu\epsilon_i^2}{(r-\theta)^2}
$$

$$
\text{s.t.} \quad \gamma_h(\boldsymbol{x}_i, y_i) \geq r - \theta - \xi_i
$$

$$
\gamma_h(\boldsymbol{x}_i, y_i) \leq r + \theta + \epsilon_i, \forall i.
$$

In our paper, we use the linear approximation (Elsayed et al., 2018) to normalize the magnitude of the norm of weights, so we can just transform this optimization target to a loss function as:

$$
\ell_{r,\theta,\mu}(\gamma_h) = \begin{cases} \frac{(r-\theta-\gamma_h)^2}{(r-\theta)^2} & \gamma_h \leq r - \theta \\ 0 & r - \theta < \gamma_h \leq r + \theta \\ \frac{\mu(\gamma_h - r - \theta)^2}{(r+\theta)^2} & \gamma_h > r + \theta, \end{cases}
$$

There is always some noise in the actual data, when deep network try to fit these data, the performance of model get worse with a larger margin variance. So we hope the larger side of the margin mean has a larger loss, which can effectively control the noise-fitting ability of models. That is why we adapt the smaller side of the margin mean to hinge form as:

$$
\ell_{r,\theta,\mu}(\gamma_h) = \begin{cases} \frac{r-\theta-\gamma_h}{r-\theta} & \gamma_h \leq r - \theta \\ 0 & r - \theta < \gamma_h \leq r + \theta \\ \frac{\mu(\gamma_h - r - \theta)^2}{(r+\theta)^2} & \gamma_h > r + \theta \end{cases}
$$

## B  PROOF OF LEMMA 2 AND THEOREM 1

**Proof of Lemma 2.**

Let $\widehat{f}_{\boldsymbol{w}}^{L-i}(\cdot) = \boldsymbol{W}_L\phi(\boldsymbol{W}_{L-1}\phi\dots\boldsymbol{W}_{i+1}\phi(\cdot))$ and $f_{\boldsymbol{w}}^{i-1}(\cdot) = \boldsymbol{W}_{i-1}\phi(\boldsymbol{W}_{i-2}\phi(\cdots\boldsymbol{W}_1(\cdot)))$, we will write the network as $f_{\boldsymbol{w}} = \widehat{f}_{\boldsymbol{w}}^{L-i}(\boldsymbol{W}_i\phi(f_{\boldsymbol{w}}^{i-1}(\boldsymbol{x})))$. If we just give $i^{th}$ layer parameter weights $\boldsymbol{W}_i$ a perturbation $\boldsymbol{U}_i$, we can have following:

$$\begin{aligned}
\|\Delta_i\|_2 &= \|(\boldsymbol{W}_i + \boldsymbol{U}_i)\phi(f_{\boldsymbol{w}}^{i-1}(\boldsymbol{x})) - \boldsymbol{W}_i\phi(f_{\boldsymbol{w}}^{i-1}(\boldsymbol{x}))\|_2 \\
&= \|\boldsymbol{U}_i\phi(f_{\boldsymbol{w}}^{i-1}(\boldsymbol{x}))\|_2 \\
&\leq \|\boldsymbol{U}_i\|_2 \cdot \|\phi f_{\boldsymbol{w}}^{i-1}(\boldsymbol{x})\|_2 \\
&\simeq \|f_{\boldsymbol{w}}^i(\boldsymbol{x})\|_2 \cdot \frac{\|\boldsymbol{U}_i\|_2}{\|\boldsymbol{W}_i\|_2}
\end{aligned} \tag{4}$$

In the last Approximate equation in Equation 4, we assume that the perturbation $\boldsymbol{U}_i$ is in the linear space span by $\boldsymbol{W}_i$, therefore, the part of $\phi(f_{\boldsymbol{w}}^{i-1}(\boldsymbol{x}))$ that is orthogonal to the space of $\boldsymbol{W}_i$ will not affect the output of perturbation. In other word, we equal the projection on the linear space of $\boldsymbol{W}_i + \boldsymbol{U}_i$ with the one on the linear space of $\boldsymbol{W}_i$, ie. $\|\phi f_{\boldsymbol{w}}^{i-1}(\boldsymbol{x})\|_2 = \frac{\|f_{\boldsymbol{w}}^i(\boldsymbol{x})\|_2}{\|\boldsymbol{W}_i\|_2}$.

$$\begin{aligned}
&\|M^{i,i+1}(f_{\boldsymbol{w}}^i(\boldsymbol{x}) + \Delta_i) - J_{f_{\boldsymbol{w}}^i(\boldsymbol{x})}^{i,i+1}(f_{\boldsymbol{w}}^i(\boldsymbol{x}) + \Delta_i)\|_2 \\
&\leq \|\boldsymbol{W}_{i+1}\|_2\|\Delta_i\|_2 \leq \frac{\|\Delta_i\|_2\|\boldsymbol{W}_{i+1}\phi f_{\boldsymbol{w}}^i(\boldsymbol{x})\|_2}{\|f_{\boldsymbol{w}}^i(\boldsymbol{x})\|_2} \cdot \frac{\|\boldsymbol{W}_{i+1}\|_2\|f_{\boldsymbol{w}}^i(\boldsymbol{x})\|_2}{\|\boldsymbol{W}_{i+1}\phi f_{\boldsymbol{w}}^i(\boldsymbol{x})\|_2} \\
&\leq \frac{\|\Delta_i\|_2\|\boldsymbol{W}_{i+1}\phi f_{\boldsymbol{w}}^i(\boldsymbol{x})\|_2}{\|f_{\boldsymbol{w}}^i(\boldsymbol{x})\|_2} \cdot \frac{\|\boldsymbol{W}_{i+1}\|_2\|f_{\boldsymbol{w}}^i(\boldsymbol{x})\|_2}{\mu_{i+1}\|\boldsymbol{W}_{i+1}\|_F\|\phi f_{\boldsymbol{w}}^i(\boldsymbol{x})\|_2} \qquad \text{Layer Cushion} \\
&\leq \frac{\|\Delta_i\|_2\|\boldsymbol{W}_{i+1}\phi f_{\boldsymbol{w}}^i(\boldsymbol{x})\|_2}{\|f_{\boldsymbol{w}}^i(\boldsymbol{x})\|_2} \cdot \frac{c\|\boldsymbol{W}_{i+1}\|_2}{\mu_{i+1}\|\boldsymbol{W}_{i+1}\|_F} \qquad \text{Activation Contraction} \\
&= \frac{\|\Delta_i\|_2\|\boldsymbol{W}_{i+1}\phi f_{\boldsymbol{w}}^i(\boldsymbol{x})\|_2}{\|f_{\boldsymbol{w}}^i(\boldsymbol{x})\|_2} \cdot \frac{c}{\mu_{i+1}r_{i+1}} \\
&\leq \frac{\|\boldsymbol{U}_i\|_2}{\|\boldsymbol{W}_i\|_2}\|f_{\boldsymbol{w}}^{i+1}(\boldsymbol{x})\|_2 \cdot \frac{c}{\mu_{i+1}r_i}
\end{aligned}$$

where $r_{i+1}$ is the stable rank of layer $i + 1$, i.e. $\frac{\|\boldsymbol{W}_{i+1}\|_F}{\|\boldsymbol{W}_{i+1}\|_2}$. Therefore by induction method we have:

$$\begin{aligned}
\|M^{i,j}(f_{\boldsymbol{w}}^i(\boldsymbol{x}) + \Delta_i) - J_{f_{\boldsymbol{w}}^i(\boldsymbol{x})}^{i,j}(f_{\boldsymbol{w}}^i(\boldsymbol{x}) + \Delta_i)\|_2 &\leq \frac{\|\boldsymbol{U}_i\|_2}{\|\boldsymbol{W}_i\|_2}\|f_{\boldsymbol{w}}^j(\boldsymbol{x})\|_2 \cdot \prod_{k=i}^{j}\frac{c}{\mu_{k+1}r_{k+1}} \\
&= \mathcal{O}(\frac{c^{j-i}}{\mu^{j-i}})
\end{aligned}$$

Obviously, we can know that $\|M^{i,j}(f^i_{\boldsymbol{w}}(\boldsymbol{x}) + \Delta_i) - J^{i,j}_{f_{\boldsymbol{w}}(\boldsymbol{x})}(f^i_{\boldsymbol{w}}(\boldsymbol{x}) + \Delta_i)\|_2 \leq \mathcal{O}(\frac{c^d}{\mu^d})$, when $d = j - i \geq 2$

$$\widehat{f}^{L-i}_{\boldsymbol{w}}(f^i_{\boldsymbol{w}}(\boldsymbol{x}) + \Delta_i) - \widehat{f}^{L-i}_{\boldsymbol{w}}(f^i_{\boldsymbol{w}}(\boldsymbol{x}))$$

$$= \|M^{i,L}(f^i_{\boldsymbol{w}}(\boldsymbol{x}) + \Delta_i) - M^{i,L}(f^i_{\boldsymbol{w}}(\boldsymbol{x}))\|_2$$

$$= \|M^{i,L}(f^i_{\boldsymbol{w}}(\boldsymbol{x}) + \Delta_i) - M^{i,L}(f^i_{\boldsymbol{w}}(\boldsymbol{x})) + J^{i,L}_{f^i_{\boldsymbol{w}}(\boldsymbol{x})}(\Delta_i) - J^{i,L}_{f^i_{\boldsymbol{w}}(\boldsymbol{x})}(\Delta_i)\|_2$$

$$\leq \|J^{i,L}_{f^i_{\boldsymbol{w}}(\boldsymbol{x})}(\Delta_i)\|_2 + \|M^{i,L}(f^i_{\boldsymbol{w}}(\boldsymbol{x}) + \Delta_i) - M^{i,L}(f^i_{\boldsymbol{w}}(\boldsymbol{x})) - J^{i,L}_{f^i_{\boldsymbol{w}}(\boldsymbol{x})}(\Delta_i)\|_2$$

$$\leq \|J^{i,L}_{f^i_{\boldsymbol{w}}(\boldsymbol{x})}(\Delta_i)\|_2 + \|(M^{i,L} - J^{i,L}_{f^i_{\boldsymbol{w}}(\boldsymbol{x})})(f^i_{\boldsymbol{w}}(\boldsymbol{x}) + \Delta_i)\|_2 + \|(M^{i,L} - J^{i,L}_{f^i_{\boldsymbol{w}}(\boldsymbol{x})})(f^i_{\boldsymbol{w}}(\boldsymbol{x}))\|_2$$

$$\leq \|J^{i,L}_{f^i_{\boldsymbol{w}}(\boldsymbol{x})}(\Delta_i)\|_2 + \mathcal{O}(\frac{c^{L-i}}{\mu^{L-i}})$$

$$\leq \|J^{i,L}_{f^i_{\boldsymbol{w}}(\boldsymbol{x})}\|_F \|\boldsymbol{W}_i\|_F \|f^{i-1}_{\boldsymbol{w}}(\boldsymbol{x})\|_2 + \mathcal{O}(\frac{c^{L-i}}{\mu^{L-i}})$$

$$\leq c\|J^{i,L}_{f^i_{\boldsymbol{w}}(\boldsymbol{x})}\|_F \|\boldsymbol{W}_i\|_F \|\phi f^{i-1}_{\boldsymbol{w}}(\boldsymbol{x})\|_2 + \mathcal{O}(\frac{c^{L-i}}{\mu^{L-i}}) \qquad \text{Activation Contraction}$$

$$\leq \frac{c}{\mu_i}\|J^{i,L}_{f^i_{\boldsymbol{w}}(\boldsymbol{x})}\|_F \|\boldsymbol{W}_i \phi f^{i-1}_{\boldsymbol{w}}(\boldsymbol{x})\|_2 + \mathcal{O}(\frac{c^{L-i}}{\mu^{L-i}}) \qquad \text{Layer Cushion}$$

$$= \frac{c}{\mu_i}\|J^{i,L}_{f^i_{\boldsymbol{w}}(\boldsymbol{x})}\|_F \|f^i_{\boldsymbol{w}}(\boldsymbol{x})\|_2 + \mathcal{O}(\frac{c^{L-i}}{\mu^{L-i}})$$

$$\leq \frac{c}{\mu_i \mu_{i\to}}|f_{\boldsymbol{w}}(\boldsymbol{x})|_2 + \mathcal{O}(\frac{c^{L-i}}{\mu^{L-i}}) \qquad \text{Interlayer Cushion}$$

$$\leq \mathcal{O}\left((r + \theta)\frac{\|\boldsymbol{U}_i\|_2}{\|\boldsymbol{W}_i\|_2}\frac{c}{\mu_i \mu_{i\to}}\right)$$

Suppose that all the perturbations $\boldsymbol{U}_i$ are independent from each other, so we can just add the influence linearly for union bound:

$$|f_{\boldsymbol{w}+\boldsymbol{u}}(\boldsymbol{x}) - f_{\boldsymbol{w}}(\boldsymbol{x})|_2 \leq \mathcal{O}\left(c(r + \theta)\sum_{i=1}^{L}\frac{\|\boldsymbol{U}_i\|_2}{\|\boldsymbol{W}_i\|_2 \mu_i \mu_{i\to}}\right).$$

**Proof. of Theorem 1.**

The proof involves chiefly two steps. In the first step we bound the maximum value of perturbation of parameters to satisfied the condition that the change of output restricted by hyper-parameter of margin $r$, using Lemma 2. In the second step we proof the final margin generalization bound through Lemma 1 with the value of KL term calculated based on the bound in the first step.

Let $\beta = \left(\prod_{i=1}^{L}\|\boldsymbol{W}_i\|_2\right)^{\frac{1}{L}}$ and consider a network structured by normalized weights $\widetilde{\boldsymbol{W}}_i = \frac{\beta}{\|\boldsymbol{W}_i\|_2}\boldsymbol{W}_i$. Due to the homogeneity of the ReLU, we have that for feedforward networks with ReLU activations $f_{\widetilde{\boldsymbol{w}}} = f_{\boldsymbol{w}}$, so the empirical and expected loss is the same for $\widetilde{\boldsymbol{w}}$ and $\boldsymbol{w}$. Furthermore, we can also get that $\prod_{i=1}^{L}\|\boldsymbol{W}_i\|_2 = \prod_{i=1}^{L}\|\widetilde{\boldsymbol{W}}_i\|_2$ and $\frac{\|\boldsymbol{W}_i\|_F}{\|\boldsymbol{W}_i\|_2} = \frac{\|\widetilde{\boldsymbol{W}}_i\|_F}{\|\widetilde{\boldsymbol{W}}_i\|_2}$. Hence, we can just assume that the spectral norm is equal across the layers, i.e. for any layer i, $\|\boldsymbol{W}_i\|_2 = \beta$.

When we choose the distribution of the prior $P$ to be $\mathcal{N}(0, \sigma \mathbf{I})$, i.e. $\boldsymbol{u} \sim \mathcal{N}(0, \sigma \mathbf{I})$, the problem is that we will set the parameter $\sigma$ according to $\beta$, which can not depend on the learned predictor $\boldsymbol{w}$ or its norm. Neyshabur et al. (2018) proposed a method that can avoid this block: they set $\sigma$ based on an approximation $\widetilde{\beta}$ on a pre-determined grid. By formalizing this method, we can establish the generalization bound for all $\boldsymbol{w}$ for which $|c_0\beta - \widetilde{\beta}| \leq \frac{1}{L}\beta$, while given a constant $c$, and ensuring that each relevant value of $c\beta$ is covered by some $\widetilde{\beta}$ on the grid, i.e. $c_1\frac{1}{\widetilde{\beta}} \leq \sum_{i=1}^{L}\frac{1}{\beta \mu_i \mu_{i\to}} \leq c_2\frac{1}{\widetilde{\beta}}$, $\mu_i \mu_{i\to}$ can be considered as a constant.

Since $\boldsymbol{u} \sim \mathcal{N}(0, \sigma^2 \mathbf{I})$, we get the following bound for the spectral norm of $\boldsymbol{U}_i$ according to the matrix extension of Hoeffding's inequalities (Tropp, 2012; Mackey et al., 2014):

$$\Pr_{\boldsymbol{U}_i \sim \mathcal{N}(0, \sigma^2 \mathbf{I})} [\|\boldsymbol{U}_i\|_2 \geq t] \leq 2\rho e^{-\frac{t^2}{2\rho\sigma^2}}. \tag{5}$$

Taking the union bound over layers, with probability $\geq \frac{1}{2}$, the spectral norm of each layer perturbation $\boldsymbol{U}_i$ is bounded by $\sigma\sqrt{2\rho \ln(4L\rho)}$. Plugging this into Lemma 2 we have that with probability $\geq \frac{1}{2}$:

$$\max_{\boldsymbol{x} \in \mathcal{X}} |f_{\boldsymbol{w}+\boldsymbol{u}}(\boldsymbol{x}) - f_{\boldsymbol{w}}(\boldsymbol{x})|_2 \leq c(r+\theta) \sum_{i=1}^{L} \frac{\|\boldsymbol{U}_i\|_2}{\beta\mu_i\mu_{i\to}}$$

$$\leq eL(r+\theta)\tilde{\beta}^{-1}\sigma\sqrt{2\rho\ln(4L\rho)} \leq \frac{r-\theta}{4}$$

We can derive $\sigma = \frac{r-\theta}{cL(r+\theta)\tilde{\beta}^{-1}\sqrt{\rho\ln(4\rho L)}}$ from the above inequality. Naturally, we can calculate the KL-diversity in Lemma 1 with the chosen distributions for $P \sim \mathcal{N}(0, \sigma^2 \mathbf{I})$.

$$D_{\mathrm{KL}}(\boldsymbol{w}+\boldsymbol{u}\|P) \leq \frac{|\boldsymbol{w}|^2}{2\sigma^2} \leq \mathcal{O}\left(c^2 L^2 \rho \ln(L\rho)\frac{(r+\theta)^2}{(r-\theta)^2}\tilde{\beta}^{-2}\sum_{i=1}^{L}\|\boldsymbol{W}_i\|_F^2\right)$$

$$\leq \mathcal{O}\left(c^2 L^2 \rho \ln(L\rho)\frac{(r+\theta)^2}{(r-\theta)^2}\sum_{i=1}^{L}\frac{\|\boldsymbol{W}_i\|_F^2}{\beta^2\mu_i\mu_{i\to}}\right)$$

$$\leq \mathcal{O}\left(c^2 L^2 \rho \ln(L\rho)\frac{(r+\theta)^2}{(r-\theta)^2}\sum_{i=1}^{L}\frac{\|\boldsymbol{W}_i\|_F^2}{\|\boldsymbol{W}_i\|_2^2\mu_i\mu_{i\to}}\right)$$

Hence, for any $\tilde{\beta}$, with probability $\geq 1-\delta$ and for all $\boldsymbol{w}$ such that, $|\tilde{\beta}-\beta| \leq \frac{1}{L}\beta$, we have:

$$L_0(f_{\boldsymbol{w}}) \leq \widehat{L}_{r,\theta}(f_{\boldsymbol{w}}) + \mathcal{O}\left(\sqrt{\frac{c^2 L^2 \rho \ln(L\rho)(r+\theta)^2 \sum_{i=1}^{L}\frac{\|\boldsymbol{W}_i\|_F^2}{\|\boldsymbol{W}_i\|_2^2\mu_i\mu_{i\to}} + \ln\frac{Lm}{\delta}}{(r-\theta)^2 m}}\right).$$

This proof method based on PAC-Bayesian framework has been raised by Neyshabur et al. (2018), we use this convenient tool for proofing generalization bound with our loss function which can obtain the optimal margin distribution.

