# OpenReview forum: "Optimal margin Distribution Network"
_ICLR.cc/2019/Conference_

### Official Review · AnonReviewer2 · 2018-10-30
**just seem interesting**

**Rating:** 5
**Confidence:** 4

**Review:**

The paper presents an improvement on the previous work by [Neyshabur et el, ICLR 2018].
More precisely, an emprical generalization bound is provided by using PAC-Bayesian empirical
bounds. To obtain the claimed improvement over the works [Barlett et al, NIPS 2017] and
[Neyshabur et el, ICLR 2018], the authors have paid attention carefully (by putting some
conditions) on the change of the layers (layer and interlayer cushion) as well as the activation
contraction. It is also worth noting that the paper is using a differrent loss function comparing
to [Neyshabur et el, ICLR 2018], which the author called Optimal Margin Distribution Loss.
Although the results seem interesting, the analysis is not convincible for me.
A plus point is that the paper presents interesting numerical experiments showing the promising of the approach.

Major comments:
1) The statement of the Theorem 1 is not clear:
is it just under the assumptions of the lemmas
or is it under all definitions and lemmas?
2) The proof of Theorem 1 is not clear:
 how do you get the inequality (5)?
how do you get an upper bound on the KL divergence?
 This is not trivial for me!
3) What is \rho in Theorem 1 and in Definition 2?
4) Your remark after Theorem 1 is not clear for me.
  you claim that the product is (3) is large, what if we restrict all the spectral norms equal to 1?
 a simple counter example would fit better the explanation here, I guest.

Minor comments:
1) The Lemma 1 and 2 are almost the same to Lemma 1 and 2 in [Neyshabur et el, ICLR 2018]
without precisely citations. I wonder how do you obtain your Lemma 1?
2) page3, after formula (1), your loss will first DECREASING, not "increasing".
Check the sentence "Fig. 1 shows, equation 1 will produce a linear loss increasing progressively with the margin distance...."

---

> ### Author Response · Authors · 2018-11-23
> **Thank you for your review**
>
> Thank you for your review and we will make the response issues clearly in revised version.
>
> ##The contribution in our paper##
> Our paper theoretically proves that margin distribution plays an important role in the generalization of deep learning. Specifically, we propose a well-designed loss function inspired by [Gao, AIJ 2013; Zhang, ICML 2017], and it can effectively alleviate the overfitting of deep models. To the best of our knowledge, this is the first work that introduces margin distribution into the analysis of deep learning. We notice that some recent works [Barlett, NIPS 2017; Neyshabur, ICLR 2018] have considered this problem, but they only focus on minimum margin, which is significantly different from our paper.
>
> The PAC-Bayesian framework is a convenient technique for margin theory analysis. In fact, other techniques like Rademacher complexity [Koltchinskii, TIT 2001; Koltchinskii, ANN STAT 2002; Bartlett, MACH LEARN 2002; Barlett, NIPS 2017] could also be used to derive similar results. In our paper the PAC-Bayesian technique is an analysis approach rather than our main contribution. As emphasized before, the theoretical contribution is relating the generalization gap to margin distribution.
>
> ##Major comments##
>
> Q1. “The statement of the Theorem 1 is not clear: is it just … definitions and lemmas?”:
> A. Theorem 1 is under all the lemmas and definitions in our paper.
>
> Q2. “The proof of Theorem 1 is not clear: how do you get the inequality (5)? how do you get an upper bound on the KL divergence?”:
> A. The inequality (5) is an extension of matrix version Hoeffding’s inequalities [Tropp, FOCS 2012; Mackey, ANN STAT 2014]. Note that the KL divergence is between two normal distributions with different mean but the same variance, so it can be bounded by |\vw|^2 / 2 \sigma^2.
>
> Q3. “What is \rho in Theorem 1 and in Definition 2?”:
> A. As presented at the beginning of the Section 3: “and $\rho$ be an upper bound on the number of output units in each layer”.
>
> Q4. “you claim that the product is (3) is large, what if we restrict all the spectral norms equal to 1?”:
> A. The impact of norms in deep learning is not similar to the linear models, we can’t directly normalize it to the value $1$. And the most common method to control the norm of weights is called weight decay, which tries to control the weights by a preset decay with a parameter. Because of its data-independence, the performance is not satisfied, and many more efficient methods to preventing the overfitting problem have been proposed, such as dropout and batch normalization. If we try to optimize the product of spectral norms directly as a regularization, we can find that the weights update of each layer is related to the product of spectral norms of other layers, the calculation cost for spectral norm is too large, and the weight decay does not consider this correlation.
>
> ##Minor comments
>
> Q1. “The Lemma 1 and 2 are almost the same to Lemma 1 and 2 … you obtain your Lemma 1?”:
> A. Lemma 1 is obtained in a similar way with PAC-Bayesian work [Neyshabur, ICLR 2018] and we have cited this paper in Sec. 3: “... we have to relate this PAC-Bayesian bound to the expected perturbed loss just like [Neyshabur, ICLR 2018] derive the Lemma 1 in their paper.” But for Lemma 2, we make a lot of nontrivial modification due to the introduction of margin variance.
>
> Q2. “page3, after formula (1), your loss will first DECREASING, not "increasing".”:
> A. We have refined the misleading description.
>
> The whole paper has been carefully improved.
>
> Thank you very much for your help!

---

> > ### Author Response · Authors · 2018-11-23
> > **Reference in our reply**
> >
> > Reference:
> > [Gao, AIJ 2013] Wei Gao and Zhi-Hua Zhou. On the doubt about margin explanation of boosting. Artificial Intelligence, 203:1–18, 2013.
> > [Zhang, ICML 2017] Teng Zhang and Zhi-Hua Zhou. Multi-class optimal margin distribution machine. In Proceedings of the 34th International Conference on Machine Learning, volume 70, pp. 4063–4071, 2017.
> > [Barlett, NIPS 2017] Peter L. Bartlett, Dylan J. Foster, and Matus J. Telgarsky. Spectrally-normalized margin bounds for neural networks. In Advances in Neural Information Processing Systems, pp. 6241–6250, 2017.
> > [Neyshabur, ICLR 2018] Behnam Neyshabur, Srinadh Bhojanapalli, and Nathan Srebro. A PAC-bayesian approach to spectrally-normalized margin bounds for neural networks. In International Conference on Learning Representations, 2018.
> > [Koltchinskii, TIT 2001] Vladimir Koltchinskii. Rademacher penalties and structural risk minimization. IEEE Transactions on Information Theory, 47(5):1902–1914, 2001.
> > [Koltchinskii, ANN STAT 2002] Vladmir Koltchinskii and Dmitry Panchenko. Empirical margin distributions and bounding the generalization error of combined classifiers. Annals of Statistics, 30, 2002.
> > [Bartlett, MACH LEARN 2002] Peter L. Bartlett, St´ephane Boucheron, and G´abor Lugosi. Model selection and error estimation. Machine Learning, 48:85–113, September 2002a.
> > [Tropp, FOCS 2012] Joel A Tropp. User-friendly tail bounds for sums of random matrices. Foundations of computational mathematics, 12(4):389–434, 2012.
> > [Mackey, ANN STAT 2014] Lester Mackey, Michael I. Jordan, Richard Y. Chen, Brendan Farrell, Joel A. Tropp. Matrix Concentration Inequalities via the Method of Exchangeable Pairs. The Annals of Probability, 42: 906–945. 2014.

---

### Official Review · AnonReviewer3 · 2018-11-02
**PAC-Bayesian analysis for DNNs**

**Rating:** 6
**Confidence:** 3

**Review:**

This paper presents a PAC-Bayesian bound for a margin loss.

Theorem 1 seems specific to ReLU activations. I wonder whether this theorem holds for other activations since most deep neural networks can use different activations at different layers instead of only the ReLU activation for all the layers. In Section 3, only Definition 3 is related to the activation. Can an activation satisfying Definition 3 have a similar bound to Theorem 1? Moreover, since the convolutional layer is a simplified case of the fully connected layer discussed in Section 3, does the convolutional layer simplify the bound in Theorem 1?

There are some typos in this paper.
“To derive a expected risk bound”: a -> an
“used to formalize error-resilience in Arora et al. (2018) as following:”: following: -> follows.
“the deep network from layer i to layer j”, “injected before level i”: i,j should be in the math mode.
“dependent on the network structure .” there is an additional blank space after ‘structure’.

---

> ### Author Response · Authors · 2018-11-23
> **Thank you for your review**
>
> Thank you for your review and we will make the response issues clearly in revised version.
>
> Q1. “whether this theorem holds for other activations … instead of only the ReLU activation for all the layers”:
> A. Current analysis only holds for ReLU, however, it’s not difficult to generalize to other activations since only the Lipschitz property is required.
>
> Q2. “Can an activation satisfying Definition 3 have a similar bound to Theorem 1”:
> A. Definition 3 is a necessary but not sufficient condition. If other conditions are also satisfied, a similar bound can be achieved.
>
> Q3. “does the convolutional layer simplify the bound in Theorem 1”:
> A. Of course Theorem 1 may be simplified by considering some special cases, but it’s beyond the scope of the paper.
>
> We have also fixed the typos and the whole paper has been carefully improved.
>
> Thank you very much for your help!

---

### Official Review · AnonReviewer1 · 2018-11-02
**Promising work, but an in-depth study of the handcrafted margin loss function is lacking**

**Rating:** 5
**Confidence:** 5

**Review:**

I consider that improving the generalization capability of neural networks on small dataset is an important line of research, and the method proposed here empirically provides great results.

The proposed margin loss (Equation 1) is said to be "specially adapted for accelerating the convergence velocity of networks by [the authors]". I would like this statement to be explained better, or at least backed by empirical evidence. In the current state, I consider that the paper lacks an in-depth study of the properties of this handcrafted loss. Few is said on the benefits of having both a linear behavior for points inside the margin and a quadratic loss for far points. The impact of loss hyperparameters (r, \gamma,\mu) should be discussed thoughtfully; at some points in the paper, r and \gamma are referred as margin mean and margin variance parameters, but this interpretation is not explained. Moreover, almost nothing is said about \mu.
By considering a simplified loss function, the provided PAC-Bayes generalization bound (Theorem 1) consider solely the flat loss region [r-\gamma, r+\gamma], but shed no light on the benefit of the hinge and quadratic parts. I conceive that this might be hard to study theoretically, but the authors should at least provide a empirical study of these.

The empirical experiments show great evidence that the proposed method successfully improve generalization capability of neural networks on small datasets compared to classical methods. I appreciate the Inter/intra class variance study of Tables 2 and 3. I would like the mathematical expression of the "hinge loss" and the "soft hinge loss" models to be explicitly written (it is not clear in the text if the soft hinge uses an hyperparameter). In the same spirit of my above comments, I would like to see how each loss hyperparameters impacts the results, instead of having access solely to the parameter values selected by the validation process.

Typos and minor comments:
- Abstract: "And our ODN model also outperforms the other three loss models..." Which three loss models?
- Section 3: "Specially, define L_0 as r=\theta..." I think it should be r=0
- Section 4.1: model-s => models
- Page 7 (and elsewhere): Table. 2 => Table 2
- Please specify that "Xent" stands for cross-entropy
- Figure 3: Please use larger font sizes
- Proof of Lemma 2: Equation. 4 => Equation 4

---

> ### Author Response · Authors · 2018-11-23
> **Thank you for your review**
>
> Thank you for your review and we will make the response issues clearly in revised version.
>
> Q1. “lacks an in-depth study of the properties of this handcrafted loss” and “The impact of loss hyper-parameters (r, \gamma, \mu) should be discussed thoughtfully”:
> A. We have added an extra appendix A to clearly explain the intuition behind this loss function.
>
> Q2. “the provided PAC-Bayes generalization … shed no light on the benefit of the hinge and quadratic parts”:
> A. Our paper proves that the generalization of deep learning heavily depends on the margin distribution. To optimize the margin distribution, we designed this ODN loss function. As a result, this well-designed loss function alleviates the overfitting problem of deep models efficiently. And we introduce how the hinge and quadratic part is derived from the intuition of margin distribution in appendix A.
>
>
> Q3. “would like the mathematical expression of the "hinge loss" and the "soft hinge loss" models to be explicitly written”:
> A. We have explicitly presented the "(soft) hinge loss" in section 4.1.
>
> We have also fixed the typos and the whole paper has been carefully improved.
>
> Thank you very much for your help!

---

### Public Comment · ~XINYI_LIN1 · 2018-10-13
**Minor comments：**

This is an interesting work. Considering margin distribution is more important than the minimum margin to the performance of model in the theoretical research in AdaBoost, applying this margin distribution principle to Deep Neural Network is a novel insight. Moreover, this paper provides a theoretical foundation of generalization bound, which can be used to search the value of hyper-parameters. In the experiment result, consistent with theoretical result , the DNN improves the performance on small sample learning problem, through introducing the margin variance to loss function. I read a similar thought in another paper submitted in this ICLR( R( https://openreview.net/forum?id=HJlQfnCqKX ),  ), it introduces a statistic $\hat{R}^2$ through a lot of empirical tests, to vertify the importance of the margin distribution. I think $\hat{R}^2$ is similar to the conception of margin variance in your paper. And the margin bound in your paper implement the future work in their paper, as they write: "We believe that the method developed here can be used in complementary with existing generalization bound".
Here is some suggestion to your paper:
1. Read the another paper I mentioned, and expand the related works;
2. In the Section 2, the hyper-parameter $r$ and $\theta$ representing margin mean and margin variance did not be stated, this trivial mistake makes me confused about this loss function;
3. Since the margin distribution is so important to get a better "representation", did the author consider the application of this ODN to adversary attack problems in the future work?

---

### Public Comment · (anonymous) · 2018-10-26
**Extreme similarity to a paper but without mentioning the source.**

Parts of this submission shows great similarity (and sometimes identical sentences) to the following NIPS paper:
“Large Margin Deep Networks for Classification” by Elsayed et al. (arXiv March 2018).

While the authors seem to have read the above reference and even adapted portions without much modification, they surprisingly do not mention the source. Below are some examples.

## Section 2 of this submission vs Section 3 of Elsayed’s ##

1. Near the beginning of Section 2 the submission:
“Define the decision boundary for each class pair {i, j} as: D{i,j} , {x | fi(x) = fj (x)}”.
The English text, the equation and even variable naming is *identical* to Eq(1) in Elsayed’s.
2. *Immediately* after this equation, the submission continues the same as Elsayed’s, with minor wording change:
Submission: “Constructed on this definition, the margin distance of a sample point x to the decision boundary Di,j is defined by the smallest translation of the sample point to establish the equation as:”
Elsayed’s: “Under this definition, the distance of a point x to the decision boundary D{i,j} is defined as the smallest displacement of the point that results in a score tie:”
Elsayed’s: “
3. The above sentence is followed by an equation that is the same as Eq 2 of Elsayed’s, including the naming convention (small delta, x, f_i, f_j).
4. Immediately after this:
Submission: “We present an approximation to γ by linearizing f_i”.
Elsayed’s: Above their Eq 6: “We present an approximation to d by linearizing f_i”, and then the same exact formula in Eq 7 of Elsayed’s appears in the submission (even similar notation conventions, like nabla_x).

## Section 4.1 ##

The middle of the first paragraph read is:
“there is a gradient term in the loss itself, which can make the computation expensive. To reduce computational cost, in the backpropagation step we considered the gradient term EQN2 as a constant, so that we recomputed the value of EQN1 at every forward propagation step.”
Elsayed’s Section 4.1, also middle of the first paragraph:
“....presence of gradients in the loss itself…. The backpropagation step for parameter updates requires the computation of second-order gradients. To further reduce computation cost to a manageable level…. treating the denominator EQN1 in (15) as a constant w.r.t. w for backpropagation. The value of EQN2 is recomputed at every forward propagation step.”

Also the end of this paragraph in the submission reads as:
“Furthermore, since the denominator item could be too small, which would cause numerical problem, we added an \epsilon with small value to the denominator so that clip the loss at some threshold.”
Elsayed’s closing sentence for their Section 4.1: “Finally, to improve stability when the  denominator is small, we found it beneficial to clip the loss at some threshold [denoted by \epsilon].”

---

> ### Author Response · Authors · 2018-10-29
> **Sorry for the missing of this reference, but the contributions of these two works are totally different.**
>
> Thanks for the comments. Our responses are proved below.
>
> Q1. Regarding the margin definition
> To our knowledge, the margin definition in section 2 is widely used in multi-class learning setting, just to name a few, the classic book [Mohri M. 2012] (cf. chapter 8.2), the top journal / conference papers [Crammer K. JMLR 2001], [Zhang T. ICML 2017], [Peter L. B. NIPS 2017]. Therefore, the margin definition is not proposed by Elsayed et al., and we treat it as a common knowledge. Note that Elsayed et al. also use this definition without citation.
>
> Q2. Regarding the linear approximation
> Thanks for pointing out the missing reference and we will cite Elsayed's paper in the revised version. However, we want to clarify that the novelty of our paper is irrelevant to this linear approximation. To be specific, we summarize the difference between our paper and Elsayed's paper in the next question.
>
> Q3. The contributions of our paper and Elsayed's paper are totally different
> The main contribution of Elsayed's work is to introduce the hinge loss into deep models across all layers to improve the empirical performance. In contrast, our contribution is:
> 1. Our paper proves that the margin distribution plays an important role in the generalization theory of deep learning;
> 2. For alleviating the overfitting problem in deep learning, we designed this ODN loss function to optimize the margin distribution.
> Therefore, our paper and Elsayed's work have totally different motivations, as we do theoretical analysis for the margin distribution of deep learning frameworks, while they mainly focus on the empirical improvement.
>
> [Mohri M. 2012] Mohri M, Rostamizadeh A, Talwalkar A. Foundations of machine learning[M]. MIT press, 2012.
> [Crammer K. JMLR 2001] Crammer K, Singer Y. On the algorithmic implementation of multiclass kernel-based vector machines[J]. Journal of machine learning research, 2001, 2(Dec): 265-292.
> [Zhang T. ICML 2017] Teng Zhang and Zhi-Hua Zhou. Multi-class optimal margin distribution machine. In Proceedings of the 34th International Conference on Machine Learning, volume 70, pp. 4063–4071, 2017.
> [Peter L. B. NIPS 2017] Peter L. B, Dylan J. F, and Matus J. T. Spectrally-normalized margin bounds for neural networks. In Advances in Neural Information Processing Systems, pp. 6241–6250, 2017.

---

### Meta-Review · Area_Chair1 · 2018-12-16
**A new optimal margin distribution loss with some good empirical results, yet premature**

**Confidence:** 4
**Recommendation:** Reject

**Metareview:**

The paper proposed an optimal margin distribution loss and applied PAC-Bayesian bounds that are from Sanov large deviation inequalities to give generalization error bounds for such a loss. Some interesting empirical results are shown to support the proposed method.

The majority of reviewers think the paper’s empirical results are encouraging, although still in premature stage. The theoretical analysis is a kind of being standard. After reading the authors’ response and revision, the reviewers do not change much of their opinions and think the paper better undergoes systematic further study on their proposal for big improvement.

Based on current ratings, the paper is therefore proposed to borderline lean rejection.